# Evaluation of Discrimination Performance in Case for Multiple Non-Discriminated Samples: Classification of Honeys by Fluorescent Fingerprinting

**DOI:** 10.3390/s20185351

**Published:** 2020-09-18

**Authors:** Elizaveta A. Rukosueva, Valeria A. Belikova, Ivan N. Krylov, Vladislav S. Orekhov, Evgenii V. Skorobogatov, Andrei V. Garmash, Mikhail K. Beklemishev

**Affiliations:** 1Department of Chemistry, M.V.Lomonosov Moscow State University, GSP-1, Leninskie Gory, 1–3, 119991 Moscow, Russia; liza_bible@yahoo.com (E.A.R.); ikrylov@laser.chem.msu.ru (I.N.K.); vladislav.orekhov@chemistry.msu.ru (V.S.O.); skorobogatoveg@gmail.com (E.V.S.); garmash@analyt.chem.msu.ru (A.V.G.); 2Laboratory of Multivariate Analysis and Global Modeling, Samara State Technical University, 244 Molodogvardeyskaya str., 443100 Samara, Russia; valerya.belickova@yandex.ru

**Keywords:** fluorescent fingerprinting, fluorophores, honey, discrimination, chemometrics, principal component analysis, tris(2,2′-bipyridyl)dichlororuthenium(II)

## Abstract

In this study we develop a variant of fluorescent sensor array technique based on addition of fluorophores to samples. A correct choice of fluorophores is critical for the successful application of the technique, which calls for the necessity of comparing different discrimination protocols. We used 36 honey samples from different sources to which various fluorophores were added (*tris*-(2,2′-bipyridyl) dichlororuthenium(II) (Ru(bpy)_3_^2+^), zinc(II) 8-hydroxyquinoline-5-sulfonate (8-Ox-Zn), and thiazole orange in the presence of two types of deoxyribonucleic acid). The fluorescence spectra were obtained within 400–600 nm and treated by principal component analysis (PCA). No fluorophore allowed for the discrimination of all samples. To evaluate the discrimination performance of fluorophores, we introduced crossing number (CrN) calculated as the number of mutual intersections of confidence ellipses in the PCA scores plots, and relative position (RP) characterized by the pairwise mutual location of group centers and their most distant points. CrN and RP parameters correlated with each other, with total sensitivity (TS) calculated by Mahalanobis distances, and with the overall rating based on all metrics, with coefficients of correlation over 0.7. Most of the considered parameters gave the first place in the discrimination performance to Ru(bpy)_3_^2+^ fluorophore.

## 1. Introduction

Fluorescence spectrum of a sample comprises a specific fingerprint, which can be used for classification purposes. Most fluorimetric fingerprinting classification methods use intrinsic fluorescence of samples [1,2]. A less developed approach is based on adding fluorophores to samples [3,4,5,6,7,8,9,10,11,12,13,14,15,16,17,18,19,20,21,22], which promises their better discrimination because of the interaction of fluorophore(s) with non-fluorescent components of the sample and corresponding changes in the spectrum. If more than one fluorophore is added to samples, we can talk about fluorescent sensor arrays [4,5,8,10,11,15,17,20].

A key issue in the development of the “add-a-fluorophore” strategy is selection of fluorophores, since their chemical nature determines the possibility of fluorescence quenching/dequenching by the sample components. To make a proper selection of fluorophore(s), a method of comparing the discrimination results is quite necessary. In case for incomplete discrimination, the simplest characteristic of performance is the number of discriminated groups. However, sometimes it does not reveal the difference between various added fluorophores. It is important to develop new techniques for assessing discrimination performance: effective, convenient, and at the same time simple.

In this work we are suggesting two measures of discrimination: first is calculation of the number of intersections of confidence ellipses in the principal component analysis scores plots (crossing number, CrN), and second is relative position (RP) characterizing the pairwise mutual location of group centers and their most distant points. The suggested parameters are compared with the standard metrics for the multiclass classification: total sensitivity (TS), that equals the ratio of correctly classified samples to the total number of samples, calculated using various methods (Mahalanobis distances between classes, quadratic discriminant analysis and support vector machine), and also the number of discriminated groups of samples in the PCA scores plots (No of groups).

In this study we classified 36 samples of honey of different origin. A number of examples of successful application of fluorescent fingerprinting technique to the discrimination of honeys is published [23]; honeys were identified and classified using 3D fluorescence in combination with parallel factor analysis [24,25], front-face fluorescence spectroscopy using factorial discriminant analysis [26] and cluster analysis [27], principal component and linear discriminant analysis [28]. No publications on adding fluorophores to honey have been found.

In an attempt to improve the discrimination of honeys with respect to using only their intrinsic emission, we added several fluorophores to the samples. No complete discrimination in one step was achieved with any fluorophore because of a large number of samples of similar nature. A large number of non-discriminated samples complicated the selection of the fluorophore allowing for the most complete discrimination. To solve this problem, we applied the suggested performance criteria.

## 2. Materials and Methods

### 2.1. Reagents and Fluorophores

We used the metal complexes proved efficient in the previous fingerprint studies as fluorophores for the middle-wave part of the visible spectrum [3]: zinc(II) complex with 8-hydroxyquinoline-5-sulfonic acid (8-Ox-Zn) (Scheme 1c) obtained by mixing equal volumes of aqueous solutions of Zn(II) chloride and the named acid (0.001 mol·L^−1^ each), and *tris*(2,2′-bipyridyl)dichlororuthenium(II) (Ru(bpy)_3_^2+^) (Aldrich) (Scheme 1a) dissolved in water to obtain a 0.001 mol L^−1^ solution and diluted to 1 × 10^−5^ mol·L^−1^ prior to use. Also, thiazole orange dye was used as a fluorophore emitting in the long-wave portion of the spectrum, and its fluorescence was enhanced by intercalating it [29] into deoxyribonucleic acid (DNA). DNA was obtained from two sources: a sodium salt from herring testes from Aldrich (DNA-1), and medicinal drug «Derinat» (ZAO FP “Technomedservis”), which contained 2.5 g DNA and 1 g NaCl in 1 L of water (DNA-2). DNA-1 was dissolved in water to obtain 7.2 × 10^−3^ mol·L^−1^ (counting the average molar weight of a nucleotide pair as 670 g/mol). DNA-1 and DNA-2 solutions were diluted to 7.2 × 10^−5^ mol·L^−1^ with water prior to use. Thiazole orange (Scheme 1b) was used as a 2.1 × 10^−4^ mol·L^−1^ solution in water. Other reagents were of analytical reagent grade and used as received. Millipore water (18 MΩ cm) was used to prepare all solutions.

### 2.2. Honey Samples

The honey samples (Nos 1–36) were produced in Russia between 2016 and 2018. The information on the botanical origin of honeys was obtained from the label (for the samples purchased in the local stores) or conveyed by the farmer (for the farm honey samples). The farm samples included polyfloral honeys 1, 2, 3, 4, 27, 32, 33, 34, 35, 36; clover (trefoil) honeys 14, 15, 16; fireweed (*Chamaenerion angustifolium*) honeys 17, 18, 19, 28, 31; linden (*Tilia*) honeys 20, 21, 22, 23, linden and raspberries honey 24; hippophae (*Rhamnus*) honey 25; white acacia honey 26; lavender honey 29; honey with galipot 30. The samples from the stores included polyfloral honeys 5, 6, 7, 8, 9, 12, 13, linden (*Tilia*) honey 10, and buckwheat (*Fagopyrum esculentum*) honey 11. Honeys were dissolved at 0.1 g/mL in water in 10-mL vials and stored at 4°C until analysis.

### 2.3. Procedures and Instrumentation

Aliquots of honey solutions (usually 50 μL) were placed in the wells of a 96-well polystyrene plate (Thermo Scientific Nunc F96 MicroWell, white, cat. No 136101) and one of the following fluorophore solutions (see Section 2.1) was added: Ru(bpy)_3_^2+^—50 μL; 8-Ox-Zn—100 μL; thiazole orange was mixed with DNA in different ratios. The obtained mixtures were diluted with distilled water to obtain a total volume of 330 μL. No fluorophores were added when registering the intrinsic emission of the samples. Each sample was measured in five parallel runs.

Agilent Cary Eclipse spectrofluorimeter with a microplate reader accessory was used to obtain emission spectra upon excitation at different wavelength for each fluorophore. Typically, we excited samples at 320 and 360 nm. The emission range was 400 to 800 nm, and both the excitation and emission slits were set at 10 nm.

### 2.4. Data Treatment

Complexity of choosing most efficient fluorophore is related with multidimensionality of initial data (several fluorophores, its respective data sets, included measurements of each sample from predetermined set on wide wavelengths region). To simplify the task, a dimension reduction method is applied to each dataset (principal component analysis; scores of two first principal component are used). After this each data set can be represented by a set of points on a plane (each point is corresponded one measurement) and the task might be reduced to assessment of relative position of groups of points (which corresponded one type of samples). Besides, we have proposed to use special indexes, as described below.

#### 2.4.1. Data Pretreatment

The raw spectral data were normalized, which involved the calculation of the formal degree of quenching (Q) as (I − I_0_)/I_0_, where I_0_ is the average emission intensity of the blank at the corresponding wavelength. Before application of each statistical method, the data were decomposed into two principal components.

#### 2.4.2. Principal Component Analysis (PCA)

In the data matrix for each of the 36 studied honeys (5 parallel wells for each) and 6 blanks, each row contained 301 intensities corresponding to wavelengths of the spectrum. The data matrix was centered and subjected to PCA using the Unscrambler X (CAMO Software, https://www.camo.com/unscrambler). Typically, two PCs were sufficient to reach a residual dispersion of ≤5%. The result was visualized as the scores plot (PC1 vs. PC2) constructed using Origin 8.1 spreadsheet (OriginLab Corp., https://www.originlab.com/index.aspx?go=PRODUCTS/Origin). Each experiment (an emission spectrum of a well) yielded one point in the scores plot. Different points belonging to the same sample were similarly shaped and colored in the Figures.

#### 2.4.3. Assessment of Discrimination Performance Using Confidence Ellipses

Confidence ellipses for each individual honey were constructed according to the Student’s distribution using the results of the five parallel measurements with a confidence level of 80% and plotted on the scores plots. The criterion for the separation of two honey samples was Mahalanobis distance (D_M_) value equal to 4, which corresponded to visual non-intersection of the ellipses. The D_M_ were calculated using the R language-based software.

As an easy way to characterize the discrimination performance, we counted the number of samples separated in the scores plot (the “Number of groups” parameter). The samples were considered separated if their confidence ellipses did not intersect.

Another technique involves manual counting the number of intersections of the confidence ellipses. We considered the number of such intersections as a measure of discrimination performance. The ellipses may intersect in a number of ways, for which the crossing numbers were postulated (Figure 1). If one ellipse lied completely inside another, then the number of intersections was postulated to be equal to 4 to account for the high degree of overlapping.

#### 2.4.4. Assessment of Discrimination Performance by Relative Position (RP) Index

Let us consider two classes each of which is characterized by its center and boundary. Above we described the data set by an ellipse covering 95% of samples; in practical situations, when the distribution law is unknown, we suggest using the simplest method of constructing the border: a convex polygon built on the boundary points of the group. For illustration (Figure 2), we use two classes of points marked in red and green. We distinguish three variants of relative position of the groups: separation (Figure 2a,b), intersection (Figure 2c), and merging (Figure 2d). The separation variant is characterized by the distance between the centers of mass: for example, in Figure 2a the distance is greater than in Figure 2b. When the classes are merged, their centers lie inside the intersection region, and when the classes just intersected, their centers lie outside the intersection region.

RP parameter is calculated using four key points: the center of the first group C1; the center of the second group C2; point D1, characterizing the first group scattering with respect to the location of the second group; point D2, characterizing the second group scattering relatively the location of the first group (Figure 3).

The ratio of the lengths of the segments C1D2 to C1D1 has the following property: In case of separation of groups, this ratio exceeds unity; in the case of intersection it is less than unity. In order to take into account the case of group merging (Figure 3b), the value of the scalar projection onto the vector C2C1 (or C1C2) is used instead of the length of the segment, because it includes information about finding the point D2 (D1) relative to C1 (C2) containing in the sign of the number. Accordingly, in the case of merging groups, the ratio of such projections C1D2 and C1B1 will be less than zero. Thus, to interpret the value, two easily remembered thresholds, 0 and 1, are used.

Points C1 and C2 are calculated by taking the average value; points D1 and D2 are selected from the following conditions: vector C1D1 is the projection of vector C1A1 onto C1C2, where A1 is the boundary point of the first class with the smallest projection of C2A1 onto C2C1. D2 point is chosen similarly with respect to the second class. To assess the relative position of the first group relative to the second, it is proposed to use the following RP value (formula 1):(1)$${\mathrm{RP}}_{12}=\frac{\mathrm{scalar}\;\mathrm{projection}\;\mathrm{of}\;\vec{C1A2}\;\mathrm{on}\;\vec{C1C2}}{\mathrm{scalar}\;\mathrm{projection}\;\mathrm{of}\;\vec{C1A1}\;\mathrm{on}\;\vec{C1C2}}$$

In case of separation of groups, the RP_12_ value is greater than 1. The larger is RP, the further the groups are located. In case of intersection, the RP value will be more than 0, but less than 1. A value less than 0 corresponds to the case of merging groups. This representation is intuitive and easy to use.

If the groups have a different level of significance, then the RP value can be calculated for the most significant group. If the groups have the same significance level, then use the smallest value from RP_12_ and RP_21_.

In an ideal case, when the boundaries of the groups set the boundary to only uniformly located points, the case of merger differs from the intersection by the number of points lying inside the intersection region. In case for crossing, the number of such points is greater than zero, but (approximately) less than half of the class volume. When the classes merge, more than approximately half of the points lie in the intersection area.

To extend the measure to N groups, the measure value for each pair of groups can be calculated and the average value taken (Formula 2):(2)$$\mathrm{RP}=\frac{1}{0.5N(N-1)}\sum_{i=1}^{N}\sum_{j=1,j\neq i}^{N}\min({\mathrm{RP}}_{ij},{\mathrm{RP}}_{ji})$$

#### 2.4.5. Assessment of Discrimination Performance by Standard Metrics

*Validity index, or total sensitivity*, was calculated as number of correctly identified samples from the training dataset (determined using Mahalanobis distances between samples and their classes, marking samples matching more than one class as potentially misclassified) divided by the number of all samples [30]. The cross-validation procedure was performed automatically by Unscrambler X software.

*Discriminant analysis (DA) and support vector machine (SVM) methods.* Linear discriminant analysis (LDA), as the name suggests, produces a linear decision boundary between two classes. In this paper, the distance from the center of the groups is calculated using the Mahalanobis distance (LDA-Mah). This metric is only valid if the two classes have similar variance–covariance matrices, otherwise the pooled covariance matrix would be inaccurate. Quadratic discriminant analysis (QDA) is another distance metric similar to the LDA; however, the distance to each class is calculated using the sample variance–covariance matrix of each class. The boundary produced by QDA is a quadratic curve, which may consist of two separate sections of boundary lines [31]. SVM is a classification method based on statistical learning, wherein a function that describes a hyperplane for optimal separation of classes is determined. In SVMs the classification rule is determined by only a small number of training set samples called support vectors (SVs). A kernel function is used to map from the original space to the feature space and can represent many forms, thus providing the ability to handle nonlinear classification cases [32]. In this paper, the input space was transformed into feature space by means of the linear, 3rd degree polynomial and radial basis function (RBF) kernels.

## 3. Results

### 3.1. Discrimination of Honeys by Intrinsic Fluorescence

We studied 36 samples of honey, purchased from local supermarkets or from apiary farmers. The samples had different consistency and color, supposing different spectral features. The fluorescence spectra of all samples are shown in Figure 4a. All spectra are different. For an excitation wavelength of 320 nm, the intrinsic emission of honeys was observed in the range of 400–600 nm. The peak at 420 nm, which is present in all spectra, is generated by the material of the plate and was not taken into account in calculations, so the scores plots were calculated for 430–550 nm. To assess the possibility of discriminating honeys by their intrinsic fluorescence, a plot of PC1-PC2 was constructed (Figure 4b): several samples are well discriminated, while the others represent one large group. Thus, honeys can be divided into 11 groups according to their intrinsic fluorescence: some of those are individual honeys and other contain several unseparated samples.

### 3.2. Fluorophores Used in the Discrimination of Honeys

The method of “fluorescent eye” is based on the interaction of fluorophores with the components of the samples. Previously, we studied a number of fluorophores in this method and found those showing favorable results include metal complexes: 8-Ox-Zn and Ru(bpy)_3_^2+^ [3,13]. In this study we also use a fluorescent dye thiazole orange (TO) that exhibits most intense emission being intercalated into DNA [33]. Moreover, DNA may form molecular complexes with marker compounds of the samples, bringing them to a closer proximity with the fluorophore.

#### 3.2.1. Thiazole Orange (TO) as Fluorophore

Appreciable emission of TO in the region higher than 500 nm is only observed in the presence of DNA (Figure 5). When considering the *whole emission spectrum*, the best discrimination in terms of the number of discriminated groups was achieved by only adding DNA without the dye to the honey (11 groups, Appendix A). However, adding thiazole orange to that system and considering only the *long-wave emission peak* allowed for separating the honeys into 12 groups (Figure 6; for the selection of the TO: DNA ratio, see Appendix A). These two systems (honey + DNA-2 by whole spectrum and honey + TO + DNA-2 by the long-wave peak) were chosen for further consideration; DNA-1 and DNA-2 without TO were also studied as blanks.

#### 3.2.2. Metal Complexes as Fluorophores

We studied the complexes 8-Ox-Zn and Ru(bpy)_3_^2+^ earlier and found them to be useful in solving classification problems of different samples [3]. Ru(bpy)_3_^2+^ added to all honey samples discriminates them into 14 classes (Figure 7a). The distances between the classes are significantly higher compared to using 8-Ox-Zn which discriminates the samples into 9 classes (Figure 7b).

#### 3.2.3. Reflection Photometry

Absorbance measurement is a technique alternative to fluorescence in array sensing. Some honey samples are noticeably colored, which offers a possibility to distinguish them by their own refection signal in visible region under illumination by a tungsten lamp. Samples with and without added Ru(bpy)_3_^2+^ were studied in Camag visualizer, and the resulting images were treated to obtain the scores plots (Appendix A). The number of discriminated groups did not exceed 5, which implies that photometry is inferior to fluorescence for discrimination purposes.

### 3.3. Discrimination Performance

In this study we tried to reach most complete discrimination of individual samples by adding fluorophores. When using that strategy, it is desirable to have a method of comparing fluorophores by their efficiency of discrimination. This is a challenging task when not all samples are discriminated. We counted the groups of points in the scores plots whose confidence ellipses did not intersect [14] to obtain the “number of groups” value (Table 1). However, in the case of plots similar to those shown in Figure 7 the number of groups does not give the correct information. For example, in Figure 7b many confidence ellipses intersect, for which reason these samples should be assigned to one group, while in fact many samples are discriminated pair-wise (for example, samples 11 and 22, 36 and 25 are clearly separated, though belonging to one large cluster of groups).

#### 3.3.1. Discrimination Performance Estimation Methods Suggested in This Study

As can be seen from the above, it is complicated to assess the discrimination performance when the samples are not well separated, or only a part of the samples forms individual classes. In an attempt to overcome these difficulties, we have suggested two characteristics of discrimination performance.

The first one is based on counting the crossing number (CrN) of the confidence ellipses over the entire scores plot (Figure 1, the obtained CrN values are listed in Table 1). Lesser CrN values correspond to better performance. Probably, more precise results would be obtained if we estimated the *overlap areas* of the confidence ellipses, but such calculations would require sophisticated programming work. Another parameter is relative position (RP) of groups (Figure 2 and Figure 3). Higher RP values imply better discrimination. The RP values for the studied fluorophores are also shown in Table 1.

#### 3.3.2. Estimation of Discrimination Performance by Standard Techniques

**Total sensitivity (TS).** This metric shows whether the point belongs to a given class and may serve as a measure of the classification performance. Having calculated the distances between each vector and each class, each sample was either assigned to a single class, marked as belonging to multiple classes or left unowned. A point was considered belonging to a class if the Mahalanobis distance *D*_M_ from the point to the class was less than or equal to 4. Based on that information, TS values by Mahalanobis distance were calculated (Table 1), only counting samples belonging to a single class.

**Discriminant analysis and support-vector machine.** Linear discriminant analysis (LDA) is only applicable to classes with similar-looking covariance matrices; therefore, we used the discriminant analysis methods applicable for independent covariance matrix estimation: quadratic discriminant analysis (QDA) and LDA-Mah, based on the calculation of the Mahalanobis distances between classes. The support-vector machine (SVM) method is widely used to classify objects with different predefined algorithms; we used linear, 3rd degree polynomial, and radial basis function [34]. Discrimination performance of LDA-Mah and SVM methods was evaluated by calculating total sensitivity on the training dataset as a measure of correctly identified samples (TS, Section 2.4). Before proceeding to QDA or SVM, the data were decomposed into principle components. Table 1 presents the results of the sensitivity calculation for the test data set.

## 4. Discussion

Adding fluorophores to samples proved to be an efficient way of improving the discrimination performance using fluorescent fingerprinting when the samples are not intensely fluorescent or are not well discriminated by their intrinsic fluorescence [3,4,5,6,7,8,9,10,11,12,13,14,15,16,17,18,19,20,21,22]. The discrimination performance should be quantitatively characterized. The simplest intuitive way of counting the number of discriminated groups faces difficulties in case of a large number of non-discriminated classes (Section 3.3, Figure 4b and Figure 7). A recognized measure of accuracy is total sensitivity (TS) that characterizes the percentage of correctly assigned samples. As alternative characteristics of performance, we suggested the CrN and RP parameters. The first metric can be calculated manually, provided the confidence ellipses have been constructed; the second is calculated in several steps including centers, borders, and scalar projections calculation (implementation as Matlab function is given in the Appendix A).

Addressing the problem of selection of the most efficient fluorophore, we can see that Ru(bpy)_3_^2+^ takes the top positions among other fluorophores, judging by CrN and RP parameters (CrN = 112, RP = 44), the number of groups, and many variants of TS (Table 1). Next efficient fluorophores are TO–DNA-1 and DNA-1 without the dye. The other fluorophores tend to show less favorable metrics than honey by intrinsic fluorescence, which implies that addition of fluorophores worsens the discrimination of samples.

To account for all metrics and compare the fluorophores quantitatively, we calculated the overall rating of fluorophores. To obtain the rating value, the individual places of fluorophores with respect to all metrics were summed up (Table 2). The best rating (minimum sum of places) is also with Ru(bpy)_3_^2+^.

It is also useful to know how the calculated parameters relate to each other. As can be seen from Table 1, the highest correlations with coefficients of 0.71–0.89 are between the new parameters CrN and RP, as well as between them and TS by Mahalanobis distance. The total sensitivity of a classification method obtained for the training dataset essentially describes the degree of separation between classes, with higher separation providing more freedom to the classification method in choosing the inter-class threshold, eventually allowing for perfect discrimination. Since the CrN parameter roughly estimates the class overlap (assuming the classes to be sampled from multivariate normal distributions), while the RP metric directly measures the distance between classes (accounting for within-class spread), these three metrics can be considered similar. The overall rating also shows a considerable correlation with CrN (*r* = 0.87) and RP (*r* = 0.89). Other metrics show a lower correlation with CrN and RP (*r =* 0.5–0.7). On the whole, all considered metrics show moderate correlation with each other, which implies that the best way of choosing the most efficient discrimination should involve using the greatest possible number of metrics. The proposed parameters CrN and RP have shown their efficiency and require further evaluation when assessing the discrimination performance.

## 5. Conclusions

Intrinsic fluorescence of 36 honey samples did not allow for their complete discrimination using PCA. Adding various fluorophores to these samples seemingly improved the discrimination; however, a quantitative assessment of the discrimination performance was needed. The task was complicated by the incomplete separation of classes. We used multiple metrics for evaluation of the discrimination performance, including standard ones (total sensitivity calculated by various methods, the number of discriminated groups) and the parameters suggested in this study (crossing number of confidence ellipses, CrN, and relative position of classes, RP). Those allowed for choosing one of the fluorophores (*tris*-(2,2′-bipyridyl) dichlororuthenium(II)) as the most efficient. However, the classification model developed in this study for the honey samples was only used to demonstrate the feasibility of using the suggested CrN and RP parameters for the evaluation of the discrimination efficiency. Correlation coefficients between the metrics confirmed that they are essentially independent and can be used together or in combination with standard metrics (the overall rating of fluorophores can be applied for that purpose).

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
