# Peer review of "Evaluation of Discrimination Performance in Case for Multiple Non-Discriminated Samples: Classification of Honeys by Fluorescent Fingerprinting"

_sensors, 2020, doi:10.3390/s20185351_

Round 1

Reviewer 1 Report

This paper (Manuscript ID: sensors-888837) describes the use of various types of fluorophores/systems for the discrimination of 36 honey samples from different sources. To further improve the discrimination performance, the authors introduced the crossing number (CrN) and relative position (RP) for calculation. In general, it is a good piece of work, and the manuscript is well written. However, the real problem is that this tool seems to be unable to perform this task; most of the honey samples still cannot be discriminated; major revisions were needed before publishing the paper.
(1) I think authors should try methods to explain the mechanism and selectivity. why authors use such dyes for honey discrimination, why the fluorescence changed after adding the honey to dyes.
(2) Page 6 (3.1), the authors have tested the intrinsic fluorescence (emission data) for honey discrimination, 11/36 samples can be discriminated. How about testing the absorption data of 36 honey samples? Maybe the results are even better.
(3) The blind test should be performed. Unknown samples should be tested to verify the accuracy of the sensing system.
(4) the sensing system should be further improved as the discrimination accuracy is still low. More 0ptimization experiments should be performed.

Reviewer 2 Report

This article describes a careful, rigorous work, the following comment from the perspective of a chemical biologist with fluorescence sensors. I do not have sufficient expertise in statistical analysis to comment in detail on your work.

This report is a similar approach reported by the same group earlier (Microchem. J., 2019, 145, 397–405) with a change in the sample from whiskey to honey. The current fingerprint strategy seems decent; however, before publication, substantial improvement will be required.

-The article lacks a clear presentation, considering the sensor point of view and describes other fluorescence results. What about the absorption behavior?

-I do not understand the importance of figures 1, 2, and 3 in the main text.

-Provide the nucleic acid sequences.

-Mention the structure of the compound used in the study.

-The fingerprint strategy is based on the interaction between the fluorophore and the sample. The author should explain what kind of interaction is possible which causes a change in fluorescence behavior (just not saying the closer proximity with the fluorophore). The authors should also provide the fluorescence graph of each group of samples in the supporting information.

-Figure 5, 1-0, should be 1:0 like that.

Round 2

Reviewer 2 Report

The authors successfully did their efforts to address my concern and improve the manuscript. 

However, I think they should correct the fig 5 legend, as it describe the molar ratio and reader may be confuse with "1-0" instead of "1:0".